# Integrating Spatio-Temporal and Generative Adversarial Networks for Enhanced Nowcasting Performance

Wenbin Yu [1,2,3], Suxun Wang [1,3,4], Chengjun Zhang [3,4], Yadang Chen [4], Xinyu Sheng [3,4,5], Yu Yao [1,3,4], Jie Liu [2,6,*] and Gaoping Liu [2,6]

1   School of Software, Nanjing University of Information Science and Technology, Nanjing 210044, China; ywb@nuist.edu.cn (W.Y.); 202212490258@nuist.edu.cn (S.W.); 202212490762@nuist.edu.cn (Y.Y.)
2   Huaihe River Basin Meteorological Center, Hefei 230031, China; liugaoping@amo.org.cn
3   Jiangsu Collaborative Innovation Center of Atmospheric Environment and Equipment Technology (CICAEET), Nanjing University of Information Science and Technology, Nanjing 210044, China; zcj@nuist.edu.cn (C.Z.); 20201249467@nuist.edu.cn (X.S.)
4   School of Computer Science, Nanjing University of Information Science and Technology, Nanjing 210044, China; adamchen@nuist.edu.cn
5   BYD Company Limited, Shenzhen 518119, China
6   Anhui Meteorological Observatory, Hefei 230031, China
*   Correspondence: liujie@amo.org.cn

**Abstract:** Nowcasting has emerged as a critical foundation for services including heavy rain alerts and public transportation management. Although widely used for short-term forecasting, models such as TrajGRU and PredRNN exhibit limitations in predicting low-intensity rainfall and low temporal resolution, resulting in suboptimal performance during infrequent heavy rainfall events. To tackle these challenges, we introduce a spatio-temporal sequence and generative adversarial network model for short-term precipitation forecasting based on radar data. By enhancing the ConvLSTM model with a pre-trained TransGAN generator, we improve feature resolution. We first assessed the model's performance on the Moving MNIST dataset and subsequently validated it on the HKO-7 dataset. Employing metrics such as Mean Squared Error (MSE), Mean Absolute Error (MAE), Structural Similarity Index Measure (SSIM), Critical Success Index (CSI), Probability of Detection (POD), and False Alarm Ratio (FAR), we compare our model's performance to existing models. Experimental results reveal that our proposed ConvLSTM-TransGAN model effectively captures weather system evolution and surpasses the performance of other traditional models.

**Keywords:** nowcasting; generative adversarial networks; deep learning; spatio-temporal sequence

## 1. Introduction

Nowcasting involves providing extremely short-range (e.g., 0–6 h) rainfall intensity predictions for local areas using radar echo maps, rain gauges, other observational data, and Numerical Weather Prediction (NWP) [1] models. It significantly impacts people's daily lives and plays a vital role in practical applications, such as predicting road conditions for drivers, offering weather guidance for regional aviation to enhance flight safety, and issuing city-wide rainfall alerts to prevent casualties.

Global warming has led to more frequent severe rainfall events in recent years, causing considerable damage to the production and livelihoods of many countries and regions worldwide [2]. As a result, meteorological forecasting remains a central challenge and focus of modern meteorological services, with accurate rainfall prediction being a primary goal for forecasters. Since the introduction of the Nowcasting concept in the 1980s, the field has evolved for nearly half a century, witnessing numerous technological and model advancements while still leaving room for improvement. Thus, addressing the challenges within nowcasting remains essential.

Various weather forecasting methods have been employed both domestically and internationally, generally falling into three categories: experience-based forecasting, statistical precipitation forecasting methods, and radar extrapolation-based precipitation forecasting methods. With deep learning and artificial intelligence rapidly advancing, researchers have started integrating short-term forecasting with deep learning, leading to the emergence of spatiotemporal sequence prediction models. In 2015, Shi et al. [3] introduced the ConvLSTM model, a significant advancement in the field of real-time forecasting. The ConvLSTM model is specifically designed to incorporate temporal features and exhibit superior representational capabilities by simultaneously establishing temporal relationships and extracting local spatial features. In 2017, Shi et al. [4] identified a positional invariance limitation within the convolutional recurrent structure of the ConvLSTM model. This limitation poses challenges since the local features present in radar echo images change over time. To overcome this issue, they proposed the TrajGRU. By actively learning the changing trajectory information within radar images, the TrajGRU model achieved enhanced predictive performance for short-term precipitation forecasting compared with ConvLSTM. Wang et al. [5] introduced the PredRNN model, which significantly improved the capturing and prediction capabilities of radar echoes. Building upon this work, they further proposed PredRNN++ [6] in 2018 to address the issue of vanishing gradients in the prediction process and improve the model's ability to capture short-term dynamic changes and sudden events. In recent years, an array of ConvLSTM-based variants has been proposed by researchers to further enhance the capabilities of the original model. These variants include fRNN [7], RLN [8], deeprain [9], CubicLSTM [10], SA-ConvLSTM [11], HPRNN [12], and others. These models represent valuable contributions to the field, each offering unique approaches and innovations to improve upon the ConvLSTM framework.

In 2020, Sønderby et al. [13] presented the MetNET model, which leverages the attention mechanism, to achieve a remarkable spatial resolution of 1 km and a temporal resolution of 2 min within a 7–8 h timeframe in the United States. This model represents a notable advancement in real-time forecasting. In 2021, the DeepMind team introduced the DGMR [14], a radar-based deep generative model. DGMR addresses the challenges associated with challenging near-term precipitation forecasts. In 2023, Cao et al. [15] introduced MIR to enhance short-term predictions of imbalanced rainfall data. Furthermore, they proposed the TSS strategy, which adopts a curriculum learning-like approach to improve the predictive performance of the model. To tackle the smoothing effect in precipitation fields and degradation in precipitation intensity forecasting, Huang et al. [16] proposed the TSRC in 2023. This deep learning-based convolutional neural network compensates for current local clues with previous local clues during the convolution process, thereby preserving more contextual information and reducing uncertainty features in deep networks. Deep learning-based methods effectively leverage data and unveil hidden linear and nonlinear features within spatiotemporal sequence data. They are capable of dimensionality reduction and clustering of high-dimensional spatiotemporal sequences. Despite the advancements made by these models in real-time forecasting, predicting radar echo sequences remains a complex problem due to the chaotic nature of the atmosphere. This complexity presents challenges in the evolution of image sequences and the establishment of effective predictive models. Furthermore, spatiotemporal prediction models struggle to generate clear and realistic radar echo images, making it difficult to meet the fine-grained requirements of short-term precipitation forecasts.

In this paper, we propose the ConvLSTM-TransGAN model, which combines the strengths of the TransGAN [17] and ConvLSTM models. Specifically, we add the pre-trained generator of the TransGAN module to the ConvLSTM module output. This approach not only inherits traditional Generative Adversarial Networks (GAN) model benefits but also tackles high-dimensional input data issues, resulting in substantial improvements in generated image quality and diversity. Additionally, the TransGAN model is a fully non-convolutional GAN, enhancing feature resolution. The ConvLSTM model is the Long and Short Term Memory (LSTM) [18] model variant that replaces matrix multiplication op-

erations for each gate with convolution operations, enabling better capture of fundamental spatial features in multidimensional data.

To assess the effectiveness and superiority of our proposed model in spatiotemporal sequence prediction tasks, we conducted experiments on the Moving-MNIST and HKO-7 radar echo datasets. We used two deep learning model evaluation metrics (Mean Absolute Error (MAE) [19], Mean Squared Error (MSE) [20], and Structural Similarity Index Measure (SSIM) [21]) and three nowcasting evaluation metrics (Critical Success Index (CSI), Probability of Detection (POD), and False Alarm Ratio (FAR) [22]) to evaluate our model's performance. The experimental results demonstrate that our proposed model accurately describes complex motion features in radar images and achieves more precise prediction results. Moreover, the generated radar echo images using our model exhibit higher clarity and realism, indicating their potential for real-world applications.

The structure of this paper is organized as follows: In Section 2, a comprehensive literature review is presented, focusing on the relevant studies concerning the proposed model. The methodology employed for model prediction is described in detail in Section 3. In Section 4, the dataset, metrics, and experimental details utilized in this study are outlined. Section 5 offers a comparative analysis of the experimental results and draws meaningful conclusions. Finally, in Section 6, a concise summary of the research's contributions is provided.

## 2. Related Works

### 2.1. Deep Learning for Nowcasting

The rise of computer science has enabled the use of Artificial Intelligence (AI) technology for nowcasting. Several models, such as multilayer perceptrons [23] and backpropagation neural networks [24], have been proposed to enhance the accuracy of short-term precipitation forecasting. However, shallow neural networks are not effective in learning from radar data due to the variability of short-term precipitation and the complexity of radar echo images, and they are prone to overfitting problems during training. In contrast, deep learning technology leverages deep nonlinear network structures and strong feature learning capabilities to represent complex functions with fewer parameters, thus improving the prediction accuracy of short-term precipitation. Deep learning methods predominantly employ deep neural networks to extrapolate large volumes of continuous radar echo images over time to predict precipitation conditions. These neural networks possess powerful learning abilities and can establish highly nonlinear, random, and complex models to solve precipitation forecasting problems. They lack the ability to uniformly model time information and spatial features and struggle to capture long-distance spatial dependence. Furthermore, prediction models based on time-space sequences find it difficult to generate clear and realistic radar echo images, which are requirements for fine-grained nowcasting. Therefore, this article proposes the integration of a pre-trained TransGAN generator at the output end of ConvLSTM to enhance image feature resolution and better capture spatiotemporal correlations.

### 2.2. TransGAN

The Transformer model was originally developed for natural language processing and has been widely successful. As it turns out, it has also found broad applications in computer vision. Unlike Convolution Neural Network (CNN) operators [25], which have limited local receptive fields, obtaining global information typically requires multiple layers of stacking. However, as the number of layers increases, the amount of information diminishes, leading to an overemphasis on certain areas of feature extraction. The Transformer's self-attention mechanism effectively captures global information and maps it to multiple spaces, thus enhancing the model's expressive power. In this section, we construct a convolution-free GAN entirely based on the Transformer architecture. This generator enhances feature resolution while also capturing low-level textures [26,27].

TransGAN addresses two significant issues in traditional GANs. First, traditional GANs suffer from mode collapse and unstable training. Second, traditional GANs rely on convolution, which has limited local receptive fields and results in detail loss at deeper levels. Compared with the current state-of-the-art GANs, TransGAN delivers highly competitive performance. Figure 1 depicts the structure of the TransGAN model.

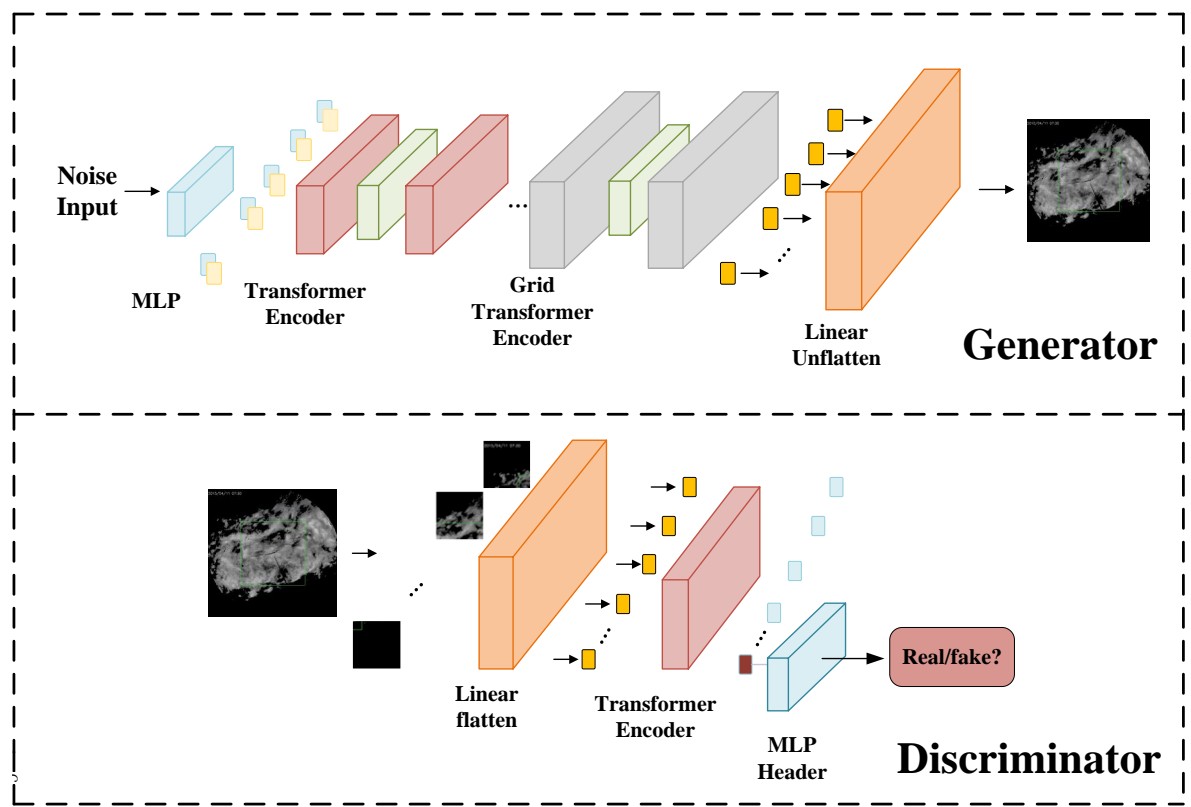

**Figure 1.** The internal structure of the TransGAN model.

### 2.3. ConvLSTM

Although LSTM, CNN, and other similar methods can capture non-linear relationships in spatiotemporal data, their effectiveness diminishes over time and cannot satisfy the requirements of collecting a large amount of spatiotemporal data. Therefore, a spatiotemporal sequence prediction model based on Convolutional LSTM was proposed in 2015, which is proficient in capturing spatial relationships and plays a vital role in spatiotemporal sequence prediction problems. This model uses convolutional operations, allowing a convolutional kernel to consider spatial positional features and extending Full-Connection Long Short-Term Memory (FC-LSTM). Consequently, it effectively addresses the data redundancy problem associated with LSTM. Figure 2 shows the internal structure of the ConvLSTM model.

In convolutional operations, the kernel size affects the speed at which motion is captured. A larger kernel size tends to capture faster motions, while a smaller kernel size tends to capture slower motions. To utilize convolutional operations, the matrix operation of FC-LSTM has been replaced. In this new approach, $i_t$, $f_t$, and $o_t$ represent the input gate, forget gate, and output gate, respectively. Additionally, $c_t$ denotes the memory cell, and $h_t$ represents the current hidden state value. The corresponding mathematical equations are provided below:

$$i_t = \sigma(\mathcal{W}_{xi} * x_t + \mathcal{W}_{hi} \times \mathcal{H}_{t-1} + \mathcal{W}_{ci} \circ C_{t-1} + b_i) \tag{1}$$

$$f_t = \sigma\left(\mathcal{W}_{xf} * x_t + \mathcal{W}_{hf} \times \mathcal{H}_{t-1} + \mathcal{W}_{cf} \circ C_{t-1} + b_f\right) \tag{2}$$

$$c_t = f_t{}^\circ c_{t-1} + i_t{}^\circ \tanh(\mathcal{W}_{xc} * x_t + \mathcal{W}_{hc} * \mathcal{H}_{t-1} + b_c) \tag{3}$$

$$o_t = \sigma(\mathcal{W}_{xo} * x_t + \mathcal{W}_{ho} \times \mathcal{H}_{t-1} + \mathcal{W}_{co}{}^\circ C_{t-1} + b_0) \tag{4}$$

$$h_t = o_t{}^\circ \tanh(c_t) \tag{5}$$

The above equations use "*" to represent the convolution operation in the calculation process, and "∘" to represent the Hadamard product. To ensure that the output and input have the same number of rows and columns, the Zero-Padding technique is applied before the convolution operation.

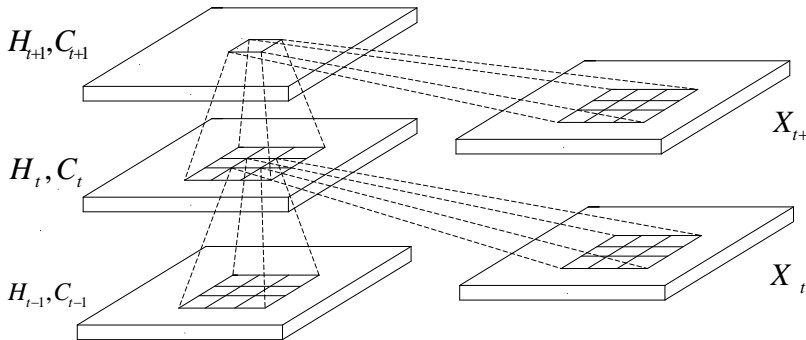

**Figure 2.** The internal structure of the ConvLSTM.

ConvLSTM can be used as a fundamental recurrent unit in encoding-decoding structures. Based on this approach, Shi et al. [3] proposed an end-to-end structure called encoding-forecasting for precipitation forecasting, as shown in Figure 3. The structure consists of two parts: the encoding network (left half) and the forecasting network (right half). ConvLSTM1 and ConvLSTM2 are encoders that encode the input image sequence into hidden variables, and ConvLSTM3 and ConvLSTM4 are decoders. All the decoders together form the forecasting network, where the current cell state and hidden variable are both copied from the output state of the encoding network. The forecasting network decodes the hidden variable and ultimately outputs the predicted image of the model. This structure can be stacked with multiple layers of ConvLSTM, resulting in a powerful nonlinear feature expression ability. It can be applied to complex spatio-temporal sequence prediction tasks, such as meteorological forecasting and traffic flow prediction.

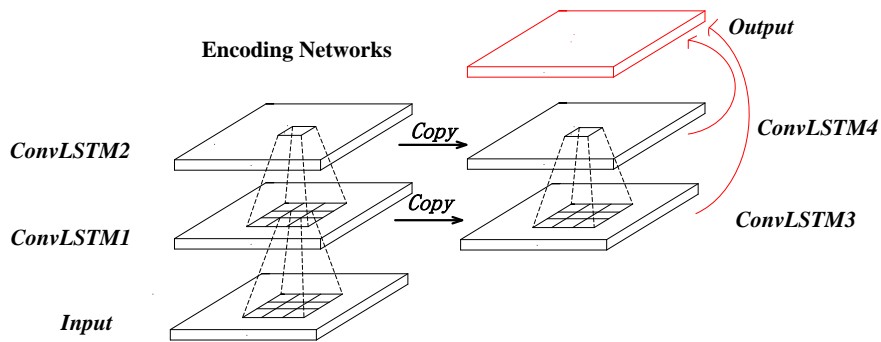

**Figure 3.** The schematic of the encoding-prediction structure based on ConvLSTM.

The entire prediction process based on encoding-decoding operations in this structure can be expressed mathematically:

$$\tilde{x}_{t+1}, \cdots, \tilde{x}_{t+K} = \underset{x_{t+1}, \cdots, x_{t+K}}{\arg\max} \; p\left(x_{t+1}, \cdots, x_{t+K} \mid \hat{x}_{t-J+1}, \hat{x}_{t-J+2}, \cdots, \hat{x}_t\right)$$

$$\approx \underset{x_{t+1}, \cdots, x_{t+K}}{\arg\max} \; p\left(x_{t+1}, \cdots, x_{t+K} \mid f_{\text{encoding}}\left(\hat{x}_{t-J+1}, \hat{x}_{t-J+2}, \cdots, \hat{x}_t\right)\right) \quad (6)$$

$$\approx g_{\text{forecasting}}\left(f_{\text{encoding}}\left(\hat{x}_{t-J+1}, \hat{x}_{t-J+2}, \cdots, \hat{x}_t\right)\right)$$

## 3. ConvLSTM-TransGAN Model

Although current models can predict more complex shape changes and trajectories in radar images, they still suffer from image blurriness over time. To address this issue, this paper proposes a ConvLSTM-TransGAN model for training radar echo sequences. This model combines the stability and generative capability of TransGAN with the feature extraction ability of ConvLSTM. Unlike traditional GAN, the TransGAN model is based on the transformer architecture and does not rely on convolutions. The generator is capable of enhancing feature resolution while capturing low-level texture details. In the following section, we will provide a detailed description of the ConvLSTM-TransGAN model.

### 3.1. TransGAN Generator

In this section, the generator of the TransGAN is introduced, as depicted in Figure 4. The process starts with the input of random noise, which is of the same size as the radar echo image used in this study, i.e., 480 × 480 × 1. The input is then transformed into a long sequence using a Multi-Layer Perceptron (MLP). The sequence is passed through the Encoder module of the Transformer to output 2D image features of size 15 × 15 × C. The image features are upsampled using bicubic interpolation to increase sampling resolution without decreasing dimensionality, resulting in 30 × 30 × C image features. These image features are reshaped into a 1D sequence, and the process of reshaping, upsampling, and reshaping is repeated twice to generate 60 × 60 × C image features. These features are then reshaped into a 1D sequence, and a pixel shuffle module replaces the bicubic interpolation to converts the 1D sequence into 120 × 120 × $\frac{C}{4}$ features.

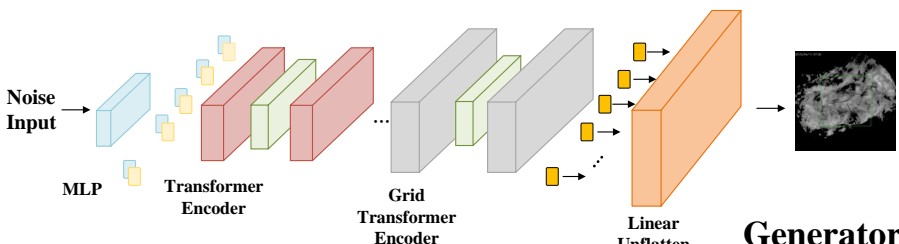

**Figure 4.** The internal structure of the TransGAN Generator.

Subsequently, a Transformer encoder block is repeated, and the resulting long sequence is reshaped into 240 × 240 × $\frac{C}{4}$. Another pixel shuffle is applied to transform the 240 × 240 × $\frac{C}{4}$ feature into a 480 × 480 × $\frac{C}{16}$ feature, followed by a linear weighting to generate the final output image of the TransGAN generator. The Transformer encoder repeats the Encoder Block L times, as shown in Figure 5. The Encoder Block comprises four parts: Layer Norm, Multi-Head Attention, Dropout, and MLP Block. This revised sentence provides a more accurate and scholarly representation of the original text.

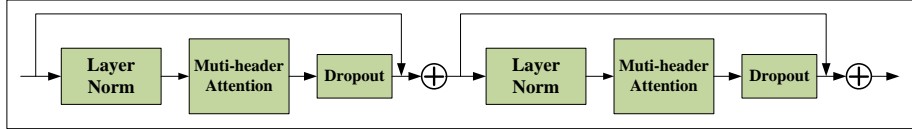

**Figure 5.** The internal structure of the Transformer Encoder.

### 3.2. TransGAN Discriminator

The discriminator component of the TransGAN still employs the Transformer Encoder as its main structure, which is illustrated in Figure 6.

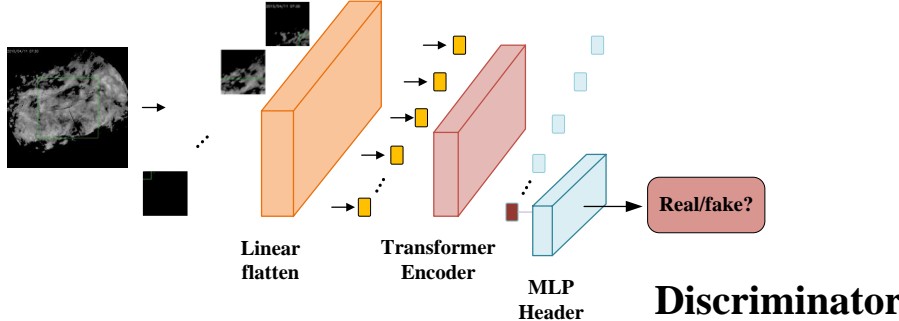

**Figure 6.** The internal structure of the TransGAN Discriminator.

For a radar echo image with an input size of $480 \times 480 \times 1$, we employ 900 $24 \times 24$ convolutional kernels with a stride of 16. This leads to a transformation represented as [480,480,1]→[30,30,576]. Furthermore, the height (H) and width (W) dimensions are flattened, which is represented as [30,30,576]→[900,576]. As the discriminator is a classification problem, before inputting it into the Transformer Encoder, a token specifically for classification is inserted into the tokens obtained earlier. This token, a trainable parameter, is a vector of length 576, similar to other tokens, and concatenated with the image. This process is represented as Cat([1,576], [900,576])→[901,576]. The output of the Transformer Encoder is of the same size as the input, and after passing through the Layer Norm layer, the classification token is extracted, and finally, the MLP Head layer is used to obtain the final output.

### 3.3. ConvLSTM-TransGAN Model

This section presents ConvLSTM-TransGAN, the final generative nowcasting model that combines the spatiotemporal feature capturing ability of ConvLSTM with the image generation capability of TransGAN. The complete structure of ConvLSTM-TransGAN is illustrated in Figure 7, where the green blocks represent the image prediction module composed of ConvLSTM's recurrent structure and the blue blocks represent the image generation module composed of the generator part of TransGAN.

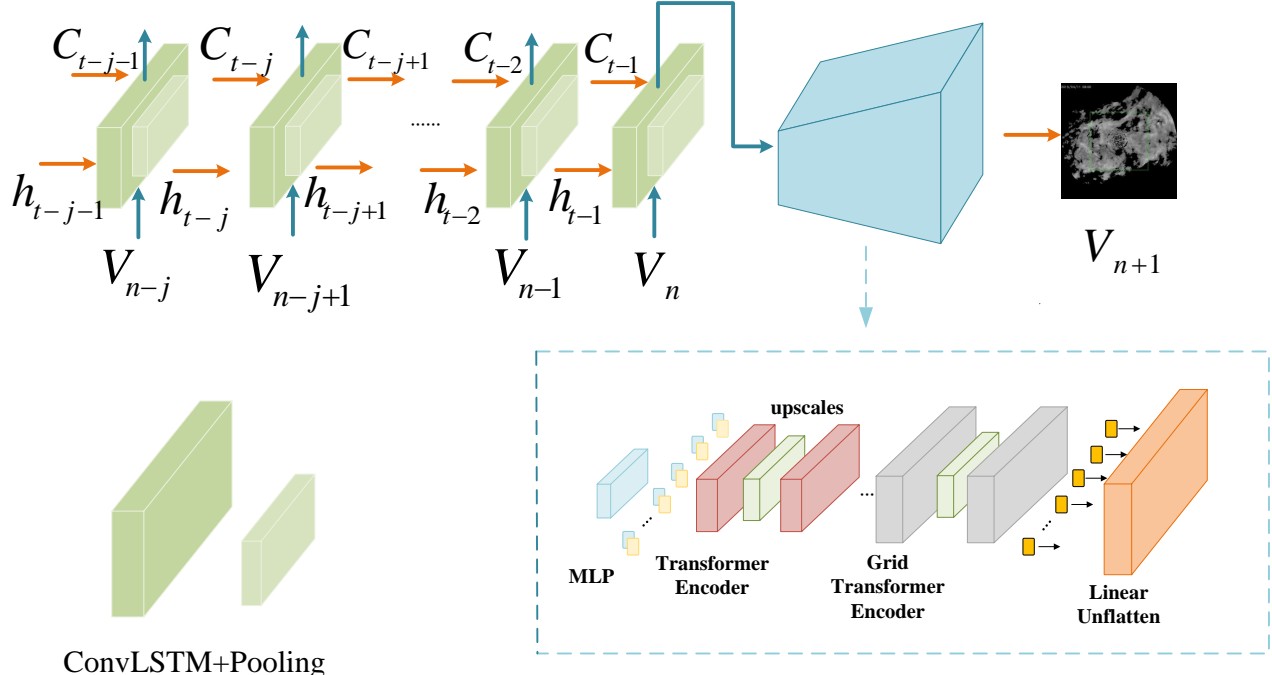

**Figure 7.** The internal structure of the ConvLSTM-TransGAN model.

The complete experimental procedure for nowcasting is presented in the following procedure:

| **The experimental procedure for nowcasting:** |
| --- |
| (a) The TransGAN is trained using real radar echo images to enable it to generate high-quality radar images. In this chapter, the HKO-7 dataset is used to train the TransGAN. |
| (b) After training the TransGAN, the generator component is extracted from TransGAN and utilized as the image generation module for ConvLSTM-TransGAN. |
| (c) The HKO-7 dataset is used as input image sequences for ConvLSTM. Through ConvLSTM's learning process, temporal information and spatial features in radar echo images are predicted. |
| (d) The output from ConvLSTM is used as input for the generator module, and with the powerful generation capability of TransGAN, clear and accurate images are obtained. |

The training process for the proposed model consists of two steps. In the first step, a radar echo image dataset is used to train the TransGAN, where the generator aims to generate new images under restricted input conditions. After the module is trained, the generator is extracted and connected to the output port of the ConvLSTM. In the second step, the same dataset used in the first step is used as the input image sequence for the ConvLSTM to learn and predict the temporal and spatial features in the radar echo images. The ConvLSTM network is then trained to generate vectors of the same dimension and constraints as the generator's input, aiming to capture the implicit features containing evolutionary information. Finally, the generator utilizes these features to produce the final prediction:

$$
\begin{aligned}
\hat{v}_{n+1} &= \arg\max_{v_{n+1}} p\left(v_{n+1} \mid \tilde{v}_{n-j+1}, \tilde{v}_{n-j+2}, \cdots, \tilde{v}_n\right) \\
&= \arg\max_{z} p\left(G(z) \mid \tilde{v}_{n-j+1}, \tilde{v}_{n-j+2}, \cdots, \tilde{v}_n\right) \\
&\approx \arg\max_{z} p\left(G(z) \mid f_{\text{convlstm}}\left(\tilde{v}_{n-j+1}, \tilde{v}_{n-j+2}, \cdots, \tilde{v}_n\right)\right) \\
&\approx G\left(f_{\text{convlstm}}\left(\tilde{v}_{n-j+1}, \tilde{v}_{n-j+2}, \cdots, \tilde{v}_n\right)\right)
\end{aligned}
\tag{7}
$$

The primary objective of the training process is to generate more realistic images rather than identify the evolution patterns of the image sequence. Directly training the ConvLSTM with images can be challenging due to the redundancy in the images, which makes it difficult to identify the underlying rules. Therefore, it is necessary to train the generator and ConvLSTM separately. The generator's performance directly determines the quality of the predicted images, which should be both low-noise and diverse. However, this poses a challenge when training GANs. The general framework of GANs proposed by Goodfellow et al. [28] is relatively coarse. To address this, we chose to use the pure transformer architecture of TransGAN. This architecture contains a generator based on a memory-friendly Transformer [29], which can gradually increase the feature resolution, and a multi-scale discriminator that can capture both semantic context and low-level texture. This architecture effectively achieves the desired performance.

## 4. Experiments

### 4.1. Datasets

#### 4.1.1. Moving MNIST

The Moving MNIST dataset is an expanded variant of the MNIST dataset extensively utilized in the field of spatiotemporal sequence prediction for ongoing research. Each dataset sequence consists of 20 successive frames, where the initial 10 frames serve as input and the last 10 frames are the desired output. Within each frame, there are two or three handwritten digits moving within a $64 \times 64$ image grid. These digits are randomly chosen from the MNIST training set, initially placed at arbitrary positions, assigned a velocity, and given a randomly selected direction from a uniform distribution on the unit circle, with the amplitude randomly chosen from the range [3, 5). When the digits converge at the same position, they rebound from the image edges and occlude each other. These unique characteristics present a challenge for the model to deliver accurate predictions without comprehending the internal dynamics of the motion. By rapidly generating real-time digits, an infinite-sized sample set can be obtained within the training set. However, due to the random digit selection, uncertain speed, and motion patterns within the Moving-MNIST dataset, if the prediction model fails to effectively extract temporal and spatial features, the resulting predictions of the model will not meet practical requirements.

#### 4.1.2. HKO-7

The Hong Kong Observatory (HKO) was established in 1883 and is situated on a small hill in the Kowloon City area of Hong Kong [30]. Hong Kong's monsoon climate and mountainous terrain contribute to highly unpredictable weather patterns and temperature fluctuations. With the continuous growth of the population and urban areas in Hong Kong, there has been an increasing demand for localized and accurate weather information. To meet this demand, HKO developed the first numerical weather forecast model in 1975 and has since accumulated a vast amount of meteorological data.

The HKO-7 dataset comprises a comprehensive collection of daily meteorological radar echo images spanning from 2009 to 2015. Each day consists of 240 frames, and each frame encompasses a resolution of $480 \times 480$ pixels, covering an extensive area of 512 km $\times$ 512 km. The original logarithmic radar reflectivity factor is transformed into pixel values using a linear equation: pixel $= \left\lfloor 255 \times \frac{dBZ+10}{70} + 0.5 \right\rfloor$, with clipped to between 0 and 255. In this context, dBZ refers to the radar reflectivity factor primarily used to describe the intensity of radar echoes. Ground clutter, sea clutter, and abnormal propagation can

introduce noise and electromagnetic interference into the original radar echo images. To achieve accurate predictions, it is necessary to perform noise reduction on the dataset.

The HKO-7 dataset contains a wealth of valuable information, and its high resolution allows for the rapid development of high-performance meteorological prediction models. These models can be further refined and enhanced through iterative training, leveraging the rich information embedded in the dataset. The availability of such a dataset enables the advancement of meteorological forecasting capabilities, contributing to more accurate and reliable weather predictions for the region.

*4.2. Datasets Processing*

In the HKO-7 dataset, radar echo data is susceptible to various interfering factors, such as ground clutter, sea clutter, and abnormal propagation, resulting in the presence of noise and electromagnetic interference in the dataset. To mitigate the impact of noise on model training and evaluation results, it is crucial to apply filtering techniques to remove these interfering factors from the dataset. Initially, one approach is to identify and remove fixed ground clutter and sun spikes from the radar images by detecting outliers' positions. For each position $i$ within the radar image boundaries, this study employs the ratio of pixel values ranging from 1 to 225 as features $x_i \sim R^{255}$. Furthermore, it estimates the sample mean $\hat{\mu}$ and covariance matrix $\hat{S}$ of these features. The computation method is as follows:

$$\hat{\mu} = \frac{\sum_{i=1}^{N} x_i}{N} \tag{8}$$

$$\hat{S} = \frac{\sum_{i=1}^{N} (x_i - \mu)(x_i - \mu)^T}{N - 1} \tag{9}$$

Furthermore, use the $\hat{\mu}$ and $\hat{S}$ to calculate the Mahalanobis distance for these features:

$$D_M(x) = \sqrt{(x - \hat{\mu})^T \hat{S}(x - \hat{\mu})^2} \tag{10}$$

Positions with a Mahalanobis distance $D_M(x)$ greater than the sum of the mean distance and three times the standard deviation are classified as outliers. After the outlier detection process, the 480 × 480 pixels in the image are divided into 177,316 inliers, 2824 outliers, and 50,260 boundary outliers [4]. Finally, to further eliminate other types of noise, pixels with values less than 71 and greater than 0 are filtered out.

*4.3. Criterions*

To comprehensively evaluate the predictive performance of the models, a set of three metrics was employed: MAE, MSE, and SSIM. MAE and MSE are widely used metrics for assessing the performance of prediction models, as they quantify the pixel-level differences between predicted and ground truth images. Smaller values of MAE and MSE indicate better agreement between the predicted and actual values. In addition to pixel-level evaluation, SSIM was utilized as a structural similarity indicator to measure the similarity between two images. SSIM goes beyond simple pixel differences and takes into account local features such as brightness, contrast, and structure, as well as their correlation and spatial information. By considering these factors, SSIM provides a more comprehensive assessment of the similarity between the predicted and ground truth images. Referring to Equations (11)–(13), it is evident that smaller values of MAE and MSE, or larger values of SSIM, indicate more accurate prediction results by the model. These metrics serve as quantitative measures to gauge the performance of the models in terms of their ability to accurately predict and reproduce the characteristics of the target data.

$$\text{MAE} = \frac{1}{n} \sum_{i=1}^{n} |\hat{y}_i - y_i| \tag{11}$$

$$\text{MSE} = \frac{1}{n} \sum_{i=1}^{n} (\hat{y}_i - y_i)^2 \tag{12}$$

$$\text{SSIM} = \frac{\left(2\mu_x\mu_y + c_1\right)\left(2\sigma_{xy} + c_2\right)}{\left(\mu_x^2 + \mu_y^2 + c_1\right)\left(\sigma_x^2 + \sigma_y^2 + c_2\right)} \tag{13}$$

In order to assess the real-time forecasting capability of the model, we utilize three commonly employed evaluation metrics in weather forecasting: CSI, POD, and FAR. To facilitate this evaluation, we transformed the forecast results and ground truth into binary matrices by applying predetermined precipitation rate echo intensity thresholds. The chosen thresholds of 0.5, 2, 5, 10, and 30 represent different levels of rainfall distribution in the HKO-7 dataset [4]. The process works as follows: if a pixel in both the predicted image and the true image is either 1 or 0, it represents a successful prediction and is recorded as True Positive (TP) or True Negative (TN). If a pixel in the true image is 1 while the corresponding pixel in the predicted image is 0, it indicates a false negative and is recorded as False Negative (FN). If a pixel in the true image is 0 while the corresponding pixel in the predicted image is 1, it indicates a false positive and is recorded as False Positive (FP). Subsequently, we calculate TP (prediction = 1, true value = 1), FN (prediction = 0, true value = 1), FP (prediction = 1, true value = 0), and TN (prediction = 0, true value = 0) using the following equations:

$$\text{CSI} = \frac{\text{TP}}{\text{FN} + \text{FP} + \text{TN}} \tag{14}$$

$$\text{POD} = \frac{\text{TP}}{\text{FP} + \text{TP}} \tag{15}$$

$$\text{FAR} = \frac{\text{FN}}{\text{FN} + \text{TP}} \tag{16}$$

The three metrics have the following meanings:

CSI: critical success index, which measures the probability of correct predictions made by the model;

POD: probability of detection, which measures the model's ability to identify rainfall;

FAR: false alarm rate, which measures the ratio of false alarms, i.e., the ratio of the model's predictions of no rainfall when there is rainfall in reality.

The CSI and POD are two evaluation metrics that reflect the accuracy of the model's predictions. The closer their values are to 1, the higher the accuracy of the model's prediction. In contrast, the FAR measures the ratio of false alarms, indicating the model's ability to predict the absence of rainfall when there is none in reality. A lower FAR value indicates stronger prediction ability and a lower probability of forecast errors. These three metrics can be used to evaluate how closely the predicted images match the real images.

### 4.4. Implementation and Training Details

In order to validate the effectiveness of our approach, we conducted an evaluation and analysis of TransGAN and commonly used GAN models. The learning rate was set to 0.0001, the patch size to 8, and the image sample size to $480 \times 480$ for all models. The loss functions employed for the generator and discriminator are presented in Table 1, and the training parameters are summarized in Table 2. It is important to note that even when working with the same dataset, different GAN models may exhibit significant variations in performance. Hence, our evaluation aimed to gain insights into the suitability of each model for the given task. Subsequently, we extracted the generator of the best-performing GAN model (TransGAN) obtained from our experiments and connected it to the output end of the ConvLSTM network, training it while comparing it to the Moving MNIST dataset. Finally, to assess the applicability of the ConvLSTM-TransGAN model in real precipitation

tasks, this chapter utilized the HKO-7 dataset as training data and replicated and compared the other four representative models: ConvLSTM, TrajGRU, PredRNN, and PredRNN++.

**Table 1.** Generator and discriminator loss functions.

| GAN | Discriminator Loss L_D (x,z;$\theta_d$,$\theta_g$) | Generator Loss L_G (z;$\theta_d$,$\theta_g$) |
|---|---|---|
| DCGAN | $-\mathbb{E}_{x \sim P_{\text{data}(x)}}\left[\log D_{\text{conv}(x)}\right] - \mathbb{E}_{z - p_z(z)}[\log(1 - D_{\text{conv}}(G_{\text{deconv}}(z)))]$ | $\mathbb{E}_{z - p_z(z)}[\log(1 - D_{\text{conv}}(G_{\text{deconv}}(z)))]$ |
| WGAN | $-\mathbb{E}_{x \sim P_{\text{data}(x)}}[D(x)] + \mathbb{E}_{z - p_z(z)}[D(G(z))]$ | $-\mathbb{E}_{z - p_z(z)}[D(G(z))]$ |
| TransGAN | $-\mathbb{E}_{x \sim P_{\text{data}(x)}}[D(x)] + \mathbb{E}_{z - p_z(z)}[D(G(z))]$ | $-\mathbb{E}_{z - p_z(z)}[D(G(z))]$ |

**Table 2.** Training parameters.

| Varient | $\alpha$ [1] | $m$ [2] | $n_D$ [3] | Optimizer |
|---|---|---|---|---|
| DCGAN | 0.0001 | 8 | 1 | Adam |
| WGAN | 0.0001 | 8 | 5 | RMSProp |
| TransGAN | 0.0001 | 8 | 1 | Adam |

[1] the learning rate. [2] the batch size. [3] the number of iterations of the discriminator per generator iteration.

## 5. Discussion

### 5.1. Comparative Analysis of Different GAN Models

In this section, we evaluate and compare the performance of three GAN models, namely DCGAN [31], WGAN [32], and TransGAN, on the same dataset. Throughout the training process, we selected four time intervals and generated four distinct samples each time, providing insights into the model's evolution.

The results depicted in Figure 8 demonstrate that all three GAN models were able to capture some of the characteristics of the radar echoes as the number of iterations increased, which indicates that they were able to learn the ground-truth to some extent. However, upon comparing the results, TransGAN demonstrated notable advantages over the other two models. Firstly, the generated images displayed diversity across the same iteration, indicating a higher degree of generalization capability of the model. Secondly, the generated images were found to be stable with minimal noise interference. Thirdly, the contrast of the images generated by TransGAN was notably higher, allowing for better identification of the precipitation regions and their intensity. Consequently, TransGAN exhibited superior generation capabilities for radar echo data in comparison to DCGAN and WGAN. Thus, in our experiments, TransGAN demonstrated the best generation performance.

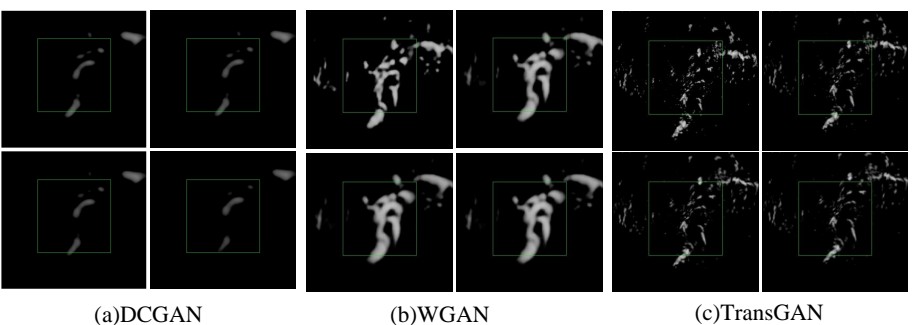

(a)DCGAN          (b)WGAN          (c)TransGAN

**Figure 8.** The radar echo predictions generated by three different Generative Adversarial Network (GAN) models.

*5.2. Qualitative Analysis of the ConvLSTM-TransGAN Model*

In this section, we present a visualization of the properties of our proposed TransGAN. Once the training of the TransGAN is completed, we extract the generator and connect it to the output end of the ConvLSTM network.

In this study, the Moving MNIST dataset was used to train the model, and for comparison purposes, the traditional models such as ConvLSTM, TrajGRU, PredRNN, and PredRNN++ were also tested. The results are presented in Figure 9, where the input is the 1–10 frame input image and the ground truth is the 11–20 frame real image.

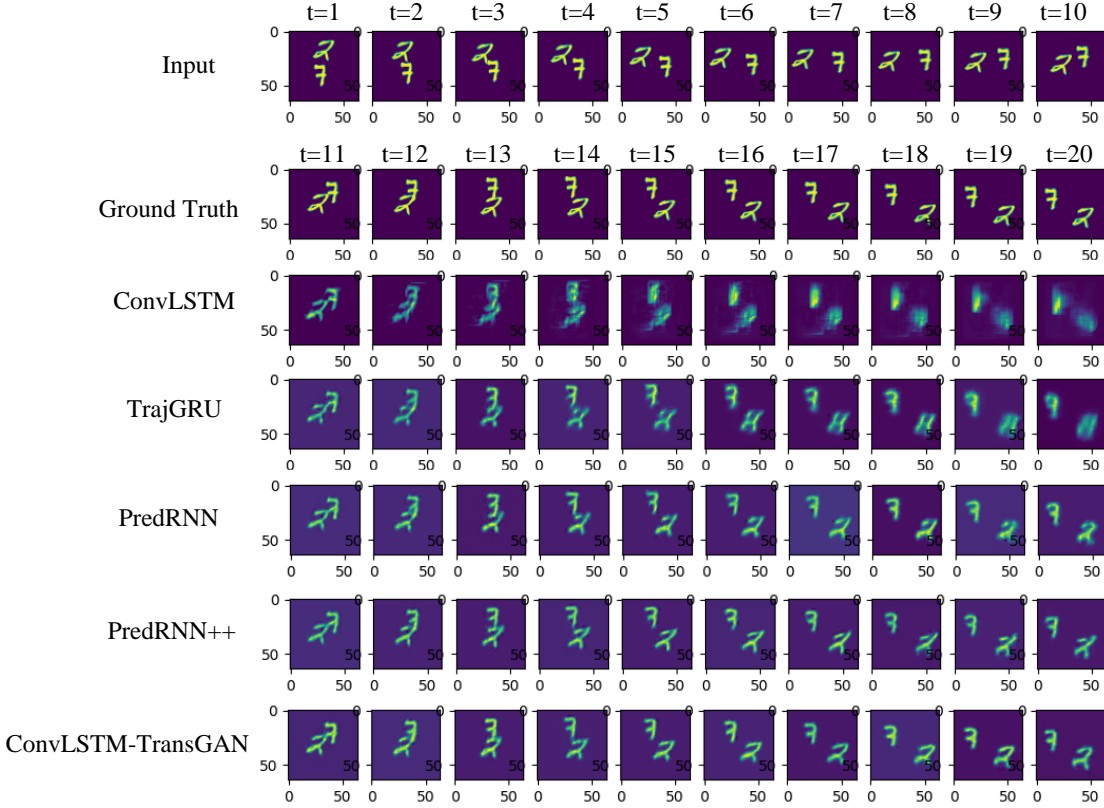

**Figure 9.** Comparison of five Models on the Moving MNIST Dataset.

The findings from Figure 9 reveal that the ConvLSTM-TransGAN model demonstrates superior performance in accurately predicting the morphological changes and motion trajectories of the digits in the image sequence, as well as producing images with higher sharpness and clarity compared with other models. Specifically, when compared with ConvLSTM, TrajGRU, PredRNN, and PredRNN++, the ConvLSTM-TransGAN model produces a more complete image of the digit "2" with higher contrast and clarity. Furthermore, ConvLSTM and TrajGRU's predicted images become increasingly blurry as time progresses, while PredRNN and PredRNN++ lose significant features in the last few frames, resulting in failed predictions of the final forms of the digits "2" and "7". In contrast, the ConvLSTM-TransGAN model accurately predicts the basic forms and motion patterns of these two digits, resulting in images that are not only clearer but also more accurate in predicting the motion trend in the image sequence.

Moreover, the model's performance was validated on the HKO-7 dataset, which comprised 812 days for training, 50 days for validation, and 131 days for testing, with data recorded every 6 min, resulting in 240 frames per day. To avoid insufficient differences between adjacent frames, the model utilized eight frames as input and predicted the next frame.

Figure 10 illustrates a performance example of the ConvLSTM-TransGAN model. The first row displays the ground truth sequence of Hong Kong's weather on 11 April 2015. This particular day was characterized by heavy rainfall, making it highly representative and typical. In this visualization, the model takes 8 frames as input and displays only the last 4 frames starting at 7:36 AM, with a frame interval of 6 min. The second row represents the actual ground-truth data. Subsequently, the third and fourth rows exhibit the predictions made by the ConvLSTM and TrajGRU models, respectively. Moving on, the fifth and sixth rows present the predictions obtained from the PredRNN and PredRNN++ models. Finally, the last row shows the prediction results of the ConvLSTM-TransGAN model. It is worth noting that the images in the last five rows are generated predictions using 8 real early frames to better demonstrate the long-term performance of the model, starting at 8:00, with a frame interval of 18 min.

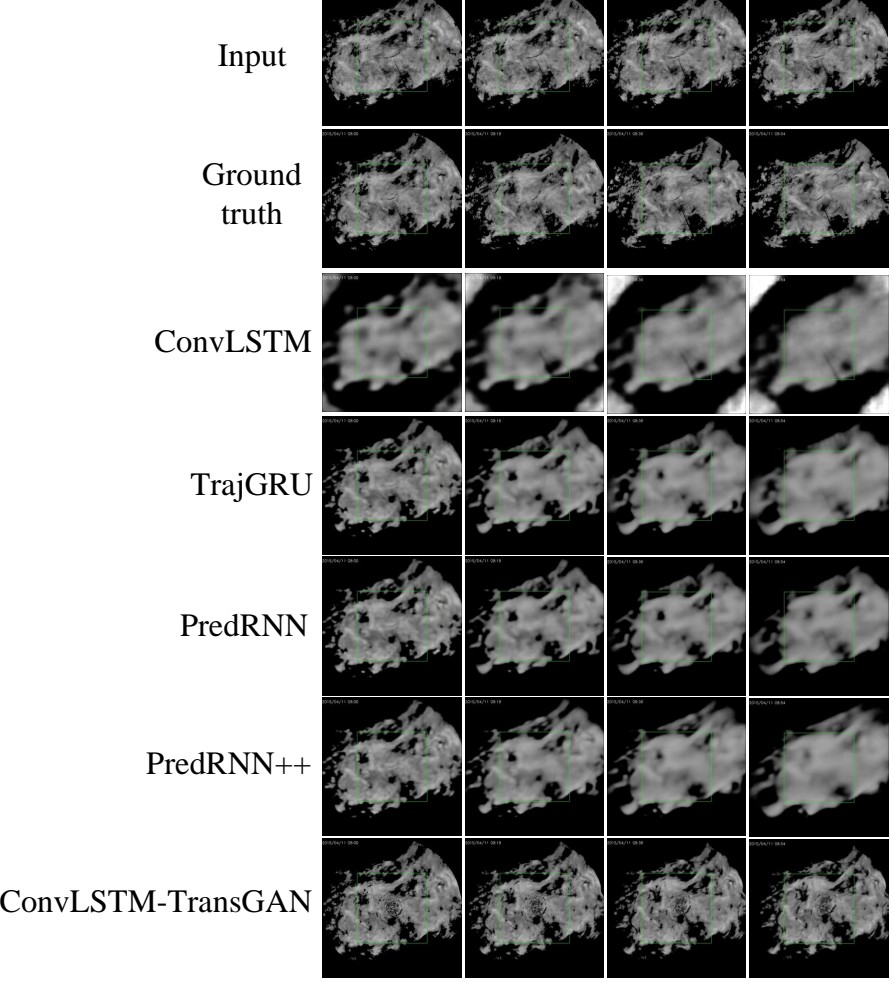

**Figure 10.** Examples for the radar echo image prediction.

Analyzing Figure 10, several observations can be made. Firstly, ConvLSTM struggles to make accurate long-term predictions for dynamic features such as the appearance and motion trajectory of radar images. On the other hand, TrajGRU, PredRNN, and PredRNN++ models show promising results in predicting the initial frames. However, as the prediction range extends to 8:36, the quality of the predicted images deteriorates, exhibiting only rough outlines and limited internal details.

In contrast, the ConvLSTM-TransGAN model demonstrates impressive performance. It accurately predicts the appearance contour of radar images and maintains consistency with the actual image in terms of motion trajectory. Notably, as time progresses, the

ConvLSTM TransGAN model produces increasingly clearer images, avoiding the problem of declining image quality observed in the TrajGRU, PredRNN, and other models. This model effectively tackles the prevalent issue of image blurriness in predictions, resulting in visually appealing results. In terms of prediction accuracy, the ConvLSTM-TransGAN model surpasses all other models in the comparison. Furthermore, it performs well in predicting extreme weather events, such as heavy precipitation, demonstrating its ability to handle typical cases effectively.

To evaluate the practicality of the ConvLSTM-TransGAN model in precipitation tasks, this chapter conducted a comparative analysis involving other four well-known models: ConvLSTM, TrajGRU, PredRNN, and PredRNN++, using the HKO-7 dataset for training. The quantitative performance of these five models was assessed using six evaluation metrics, namely MAE, MSE, SSIM, CSI, POD, and FAR, as summarized in Table 3. The values of these six metrics are the average of the last 10 frames of the prediction results and are obtained by averaging the results of multiple experiments.

The results demonstrate that the ConvLSTM-TransGAN model outperforms the other four models across all evaluation metrics, showing significant improvements. By incorporating the generative network, the ConvLSTM-TransGAN model enhances its prediction capabilities while maintaining the accuracy of spatiotemporal sequence prediction, thereby generating radar echo images of higher quality. Moreover, the ConvLSTM-TransGAN model exhibits promising performance in the three short-term precipitation prediction evaluation metrics, further validating its effectiveness.

Overall, the findings support the feasibility and superiority of the ConvLSTM-TransGAN model in practical precipitation tasks, highlighting its potential as a valuable tool for improved prediction and the generation of radar echo images.

**Table 3.** The comparative performance of various models in terms of their evaluation metrics on the HKO-7 dataset. ↑ stands for the higher, which denotes a better result. ↓ stands for the lower, which denotes a better result. Bold represents the data with the best performance in different models.

| Model | MAE ↓ | MSE ↓ | SSIM ↑ | CSI ↑ | POD ↑ | FAR ↓ |
|---|---|---|---|---|---|---|
| ConvLSTM | 716.3 | 591.1 | 0.672 | 0.689 | 0.809 | 0.284 |
| TrajGRU | 486.8 | 439.1 | 0.696 | 0.719 | 0.817 | 0.277 |
| PredRNN | 420.7 | 370.3 | 0.724 | 0.743 | 0.827 | 0.237 |
| PredRNN++ | 372.5 | 310.9 | 0.816 | 0.762 | 0.831 | 0.230 |
| ConvLSTM-TransGAN | **288.1** | **244.9** | **0.854** | **0.809** | **0.842** | **0.203** |

To visually illustrate the disparities in prediction performance among the models, this study compares the output results of each model for each frame and plots the change curves of different evaluation metrics in relation to the prediction time steps. Specifically, Figure 11 depicts the comparison using the MAE, MSE, and SSIM evaluation metrics. The findings indicate that as the prediction time steps increase, the performance of all models in terms of prediction deteriorates. However, the ConvLSTM-TransGAN model consistently maintains the highest level of performance across all evaluation metrics. This indicates that the model effectively captures spatiotemporal features while generating clear and accurate radar images. The results highlight the superiority of the ConvLSTM-TransGAN model over ConvLSTM, TrajGRU, PredRNN, and PredRNN++ in spatiotemporal sequence prediction and radar image generation. The ConvLSTM-TransGAN model stands out as it consistently outperforms the other models, reinforcing its capability to produce high-quality predictions and confirming its effectiveness in capturing complex spatiotemporal relationships.

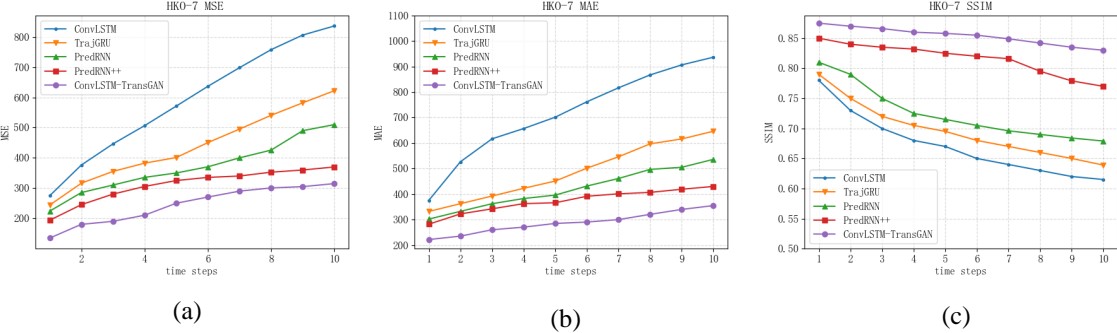

**Figure 11.** Subfigures (**a**–**c**) describe the variation curves of ConvLSTM, TrajGRU, PredRNN, PredRNN++, and ConvLSTM-TransGAN models relative to the predicted time steps using MSE, MAE, and SSIM evaluation metrics.

### 5.3. Selection of Spatio-Temporal Prediction Models

In this study, our focus was on selecting an appropriate spatio-temporal sequence model to achieve accurate nowcasting. We compared two candidate models, ConvLSTM and PredRNN, in our experiments. Specifically, we conducted comparative analyses between the ConvLSTM-TransGAN model and the PredRNN-TransGAN model, computing their MSE, MAE, and SSIM as shown in Table 4.

**Table 4.** Performance Comparison of ConvLSTM and PredRNN models under TransGAN Generative model. ↑ stands for the higher, which denotes a better result. ↓ stands for the lower, which denotes a better result.

| Model | MAE ↓ | MSE ↓ | SSIM ↑ |
|---|---|---|---|
| PredRNN-TransGAN | 287.7 | 244.6 | 0.855 |
| ConvLSTM-TransGAN | 288.1 | 244.9 | 0.854 |

From Table 4, it can be observed that the performance metrics of the two models are very close, especially in terms of SSIM. Based on this, we can conclude that the TransGAN generative module has the capability to effectively enhance the image quality of both ConvLSTM and PredRNN models during the image augmentation process. In this regard, there is not a significant difference between ConvLSTM and PredRNN.

Indeed, the ConvLSTM model holds a remarkable position as a milestone in deep learning for nowcasting, exhibiting strong representativeness. It has already demonstrated its ability to capture spatio-temporal information, and subsequent models such as PredRNN and PredRNN++ are based on improvements to the ConvLSTM framework. The TransGAN generative model effectively optimizes information extraction for ConvLSTM.

In our experiment, the addition of the TransGAN image generation module at the output of ConvLSTM and PredRNN resulted in comparable results. Compared with more complex models such as PredRNN and its improvements, the ConvLSTM is simpler and more efficient in inference performance.

Based on the aforementioned considerations, we selected the ConvLSTM model as the spatio-temporal sequence module in this study. Its representativeness, capability to capture spatio-temporal information, and advantages in model reasoning make ConvLSTM the optimal choice for achieving accurate nowcasting.

### 5.4. How Does the Learning Rate Affect the Prediction Performance?

The learning rate, denoted as $\alpha$, plays a crucial role as a hyperparameter in the field of deep learning. It governs the magnitude of parameter updates during the training process. Choosing an appropriate learning rate can expedite the convergence rate of the model, mitigate fluctuations, enhance generalization capabilities, and effectively support various optimization algorithms. Consequently, it significantly improves the overall performance of

the model. In our specific experiment, we begin by training the TransGAN image generation module until it achieves sufficient convergence and balance. Afterward, we freeze its parameters and use the trained module as the postprocessing module for the feature extraction module. Subsequently, we continue to train the feature extraction module based on the ConvLSTM architecture. During this training process, we conduct a comprehensive analysis using different $\alpha$ values, specifically 0.00001, 0.0001, 0.001, 0.01, and 0.1. After meticulously scrutinizing the results presented in Table 5, it is evident that when $\alpha$ is set to 0.0001, the indicators MSE, MAE, and SSIM all reach the optimal level, indicating that the model can effectively capture spatiotemporal features while generating clear and accurate radar images at $\alpha = 0.0001$. In contrast, when using a lower learning rate of 0.00001, the model's training process becomes hindered as it gets trapped in local optima. This phenomenon extends the training time and consequently hampers the improvement rate of the evaluation metrics, leading to suboptimal results.

**Table 5.** Performance with different $\alpha$. ↑ stands for the higher, which denotes a better result. ↓ stands for the lower, which denotes a better result. Bold indicates the data with the best performance among different Learning rate.

| Model Strategy | Training | MSE ↓ | MAE ↓ | SSIM ↑ |
|---|---|---|---|---|
| | $\alpha = 0.1$ | 754.1 | 695.2 | 0.826 |
| | $\alpha = 0.01$ | 574.6 | 542.8 | 0.833 |
| ConvLSTM-TransGAN | $\alpha = 0.001$ | 383.4 | 358.2 | 0.841 |
| | $\alpha = 0.0001$ | **288.1** | **244.9** | **0.854** |
| | $\alpha = 0.00001$ | 312.3 | 279.4 | 0.843 |

*5.5. Limitation*

In recent years, the rapid advancement of the artificial intelligence industry and deep learning technology has provided novel opportunities for the investigation of short-term and near-term precipitation forecasting. This study employed deep learning methodologies to explore the realm of short-term and near-term precipitation prediction, yielding promising prognostic outcomes. Nevertheless, there remain several aspects that warrant further inquiry.

Firstly, the current body of research predominantly relies on radar echo images as the primary data source for precipitation prediction, disregarding the multifaceted influences that affect precipitation. Such overreliance on radar echo images may introduce errors, thereby potentially compromising the accuracy of predictions.

Secondly, the extraction of long-range spatial correlation features from radar images necessitates the accumulation of information over multiple temporal intervals. However, the ConvLSTM-TransGAN model proposed in this paper adopts a sequential computation approach, progressively attenuating the efficacy of information over time. Consequently, the acquisition of long-range features becomes increasingly limited, leading to a decline in the precision of predictions. To mitigate this issue, the paper aimed to maximize the utilization of radar echo images and capture global features to the greatest extent possible. Consequently, the model exhibits a substantial parameter count and complexity, thereby augmenting the challenges associated with its training process.

**6. Conclusions and Future Work**

This paper provides a comprehensive analysis of the current state of nowcasting. We assess the limitations of traditional and deep learning-based methods and propose new models and strategies to improve the accuracy, clarity, and refinement of nowcasting. Specifically, We propose a generative nowcasting model that combines TransGAN and ConvLSTM to address issues of insufficient realism and blurry images in predicted images generated by spatiotemporal sequence prediction models. The proposed model addresses the issues of insufficient realism and blurry images in predicted images generated by spatiotemporal sequence prediction models. By leveraging the strengths of TransGAN and

ConvLSTM, our model achieves significant advantages in various evaluation metrics. It demonstrates high prediction accuracy by generating clear and realistic radar echo images and successfully tackling the problem of image blurring in long-term forecasting. These results indicate the practical and valuable application potential of our model in short-term precipitation forecasting, meeting the business requirements of nowcasting.

Overall, the proposed method is effective in meeting practical requirements for nowcasting and offers new ideas and solutions for further constructing accurate, clear, and refined models. The model has strong applicability to meteorological forecasting tasks and presents new opportunities for nowcasting research in light of recent developments in the artificial intelligence industry and deep learning technology. However, the data set used in this study is relatively limited and mainly focused on one region (Hong Kong, China), which may affect the prediction accuracy to a certain extent when dealing with complex nowcasting problems. Future research can aim to address this limitation by collecting new training samples and incorporating more meteorological data on precipitation to further improve the accuracy of prediction.

**Author Contributions:** Conceptualization, W.Y. and S.W.; data curation, W.Y. and S.W.; formal analysis, W.Y.; funding acquisition, C.Z. and J.L.; investigation, Y.C. and X.S.; methodology, W.Y. and S.W.; project administration, G.L.; resources, C.Z. and J.L.; software, X.S.; supervision, Y.C. and J.L.; validation, S.W. and X.S.; visualization, W.Y. and Y.C.; writing—original draft, S.W. and Y.Y.; writing—review and editing, W.Y., S.W. and C.Z. All authors have read and agreed to the published version of the manuscript.

**Funding:** This work is supported by the National Natural Science Foundation of China (Grant No. 61501247, No. 61703212, No. 61802197, No. 62071240), and the National Science Foundation of Jiangsu Province of China (Grant No. BK20160971, No. BK20171458).

**Data Availability Statement:** All two radar echo datasets are publicly available. Moving MNIST can be downloaded from http://www.cs.toronto.edu/~nitish/unsupervised_video (accessed on 2 May 2023). HKO-7 can be found at https://github.com/sxjscience/HKO-7 (accessed on 2 May 2023).

**Acknowledgments:** The authors thank the anonymous reviewers and the editors for their insightful comments and helpful suggestions to improve our manuscript.

**Conflicts of Interest:** The authors declare no conflict of interest.

## Abbreviations

The following abbreviations are used in this manuscript:

| | |
|---|---|
| MSE | Mean Squared Error |
| MAE | Mean Absolute Error |
| SSIM | Structural Similarity Index Measure |
| CSI | Critical Success Index |
| POD | Probability of Detection |
| FAR | False Alarm Ratio |
| NWP | Numerical Weather Prediction |
| GAN | Generative Adversarial Network |
| LSTM | Long and Short Term Memory |
| AI | artificial intelligence |
| CNN | Convolution Neural Network |
| MLP | Multi-Layer Perceptron |
| FC-LSTM | Full-Connection Long Short-Term Memory |
| HKO | Hong Kong Observatory |

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
