# Peer review of "Integrating Spatio-Temporal and Generative Adversarial Networks for Enhanced Nowcasting Performance"

_remotesensing, doi:10.3390/rs15153720_

Round 1
Author Response
Dear Reviewer,
I hope this letter finds you well. I am writing to submit a revised version of our manuscript titled "Integrating Spatio-Temporal and Generative Adversarial Networks for Enhanced Nowcasting Performance" originally submitted to Remote Sensing. We sincerely appreciate your comprehensive review and appreciate your suggestions for our work.
We are pleased to note that you have not found any objections or concerns regarding the content of our manuscript. We appreciate your positive feedback and appreciate your time and effort in reviewing our work.
As part of the revision process, we carefully reviewed the manuscript to ensure its clarity and compliance with the journal's guidelines. Although you did not provide specific suggestions for revisions, we took this opportunity to conduct final proofreading and make adjustments to enhance the readability and cohesion of the manuscript.
In addition, we have resolved any potential printing errors and inconsistencies discovered during the thorough review process. Our goal is to present our research in the simplest and most accurate way possible.
Please find attached the revised draft for your further consideration. We once again sincerely thank you for the positive evaluation of our work.
Thank you for your time and attention.
Best regards,
All authors

Reviewer 2 Report
This paper proposes a deep learning-based radar echo extrapolation model by combing ConvLSTM and TransGAN, where the ConvLSTM is used for extracting feature from previous
radar echo sequence, the TransGAN is used to generate the radar echo field based the feature extracted by ConvLSTM. The experimental results have illustrated superiority of the proposed model on predicting more accurate and realistic radar echo fields. However, there are still some concerns need to be addressed, listed as follows:
· Grammar and phrasing: There are some non-standard expressions and traces of automatic translation, including but not limited to the following:
Lines 137-138: "This description offers a more precise and academic rendition of the original paragraph." This sentence seems unrelated to the previous text and is abrupt.
Lines 460-461: "This paper provides a comprehensive analysis of the current state of nowcasting, both domestically and internationally." What do "domestically" and "internationally" refer to?
· In the introduction and related work sections, the description of current deep learning methods is relatively vague, without summarizing specific problems and how this paper solves them.
· Regarding the experiments:
(1) Analysis of typical cases, especially heavy precipitation cases, is lacking.
(2) Lines 402-403: "The quantitative performance of these five models was assessed using six evaluation metrics, namely MAE, MSE, SSIM, CSI, POD, and FAR, as summarized in Table 3." How were the results in Table 3 calculated? Were all samples combined for one calculation or were evaluations done separately for each forecast and then averaged? What were the thresholds chosen for precipitation intensity or radar reflectivity? Also, there is no evaluation result for different thresholds.
(3) Section 5.3: Does the learning rate refer to the learning rate of the ConvLSTM? Since the author proposed that the feature extraction module and image generation module are trained independently, when studying the impact of learning rate, did the authors assume that the image generation module had already been sufficiently trained?
Author Response
Dear Reviewer,
I hope this letter finds you well. I am writing to submit a revised version of our manuscript titled "Integrating Spatio-Temporal and Generative Adversarial Networks for Enhanced Nowcasting Performance" originally submitted to Remote Sensing. We sincerely appreciate your comprehensive review and appreciate your suggestions for our work.
Your constructive comments have been instrumental in improving our manuscript. You raised concerns regarding grammar and phrasing, as well as some questions related to our experiments. That really helps us a lot. We have carefully considered each of your suggestions and have made the necessary revisions to address your concerns. Below, we summarize the major changes we made in response to your feedback.
Grammar and phrasing: There are some non-standard expressions and traces of automatic translation, including but not limited to the following:
Lines 137-138: "This description offers a more precise and academic rendition of the original paragraph." This sentence seems unrelated to the previous text and is abrupt.
Lines 460-461: "This paper provides a comprehensive analysis of the current state of nowcasting, both domestically and internationally." What do "domestically" and "internationally" refer to?
Answer:Thank you for your kind suggestions. We conducted a comprehensive proofreading of the manuscript and revised sentences to enhance clarity and coherence. Ambiguous phrases were rephrased to ensure that our intended meaning is conveyed effectively.
In this article, we have made the following modifications to it, in line 139,line 534,we have made modifications or deletions to it.
In the introduction and related work sections, the description of current deep learning methods is relatively vague, without summarizing specific problems and how this paper solves them.
Answer:Thank you for your kind suggestions. In this revised draft, we have made the following modifications to it.
Lines 70-75, we supplemented and added some of the issues present in the current deep learning models.
Lines 76-85, we are dedicated to addressing these problems through our proposed model.
Regarding the experiments:
1. Analysis of typical cases, especially heavy precipitation cases, is lacking.
Answer: Thank you for your kind suggestions. In this article, the data used for Figure 10 is the weather ground truth data from the Hong Kong Observatory on April 11, 2015, specifically from 7:00 AM to 9:00 AM. On that particular day, there was an occurrence of heavy precipitation.
In this article, we have made the following modifications to it.
In lines 412-439, we provided detailed supplementary explanations related to the heavy precipitation on that day.
2. Lines 402-403: "The quantitative performance of these five models was assessed using six evaluation metrics, namely MAE, MSE, SSIM, CSI, POD, and FAR, as summarized in Table 3." How were the results in Table 3 calculated? Were all samples combined for one calculation or were evaluations done separately for each forecast and then averaged? What were the thresholds chosen for precipitation intensity or radar reflectivity? Also, there is no evaluation result for different thresholds.
Answer:Thank you for your kind suggestions. In this article, we have made the following modifications to it.
In lines 444-446, we have included the calculation method for the data presented in Table 3.
In lines 327-328, equation(11-13), lines 343-344, equation(14-16), the detailed calculation formulas are provided.
In lines 330-334, we have increased the threshold for precipitation intensity.
3. Section 5.3: Does the learning rate refer to the learning rate of the ConvLSTM? Since the author proposed that the feature extraction module and image generation module are trained independently, when studying the impact of learning rate, did the authors assume that the image generation module had already been sufficiently trained?
Answer:Thank you for your kind suggestions. In Section 5.4, the term "learning rate" refers to the learning rate used in training the ConvLSTM model. After achieving sufficient convergence and equilibrium in the image generation module, we freeze the trained parameters and utilize it as the postprocessing of the feature extraction module. Subsequently, we adjust the learning rate specifically for the feature extraction module.
In this article, we have made the following modifications to it.
In lines 500-505, We have supplemented this aspect to make it easier to understand.
We believe that these enhancements have significantly improved the quality and reliability of the manuscript. Additionally, we have carefully reviewed the manuscript for any remaining language issues and have ensured that it adheres to the journal's guidelines.
Please find attached the revised manuscript along with a point-by-point response. We sincerely appreciate the thoughtful evaluation provided by your comments, which has undoubtedly strengthened our research. We also extend our gratitude to you for time and consideration throughout this process.
Thank you for your time and attention.
Best regards,
All authors

Reviewer 3 Report
Comments to the manuscript remotesensing2484788
Integrating Spatio-Temporal and Generative Adversarial Networks for Enhanced Nowcasting Performance
by Wenbin Yu, Suxun Wang, Yadang Chen, Chengjun Zhang, Xinyu Sheng, Yu Yao, Jie Liu, Gaoping Liu
The paper is devoted to meteorological forecasting methods and unambiguously demonstrated the advantage of the ConvLSTM-TransGAN model of the authors over other four well-known models: ConvLSTM, TrajGRU, PredRNN, and PredRNN++. At the same time, the authors described in detail the limitations of their method. The paper deserves publication in the journal Remote Sensing, but it is not without flaws, noted below in the form of comments.
Main comments
1. Line 280: what is dBZ in the pixel expression?
2. Lines 306-313: is there any physical interpretation of the abbreviations TP, FN, FP, TN?
3. Line 344, Figure 8: would you like to clarify how the 4 frames of the sequence differ for a specific GAN model?
4. Line 389: Figure 10: How does this figure illustrate image prediction? Most likely, there is a good image recognition here?
5. Line 439, Table 4: what can explain the worsening of indicators for α=0.00001?
Some inaccuracies
1. Line 42: replace learnsing with learning.
2. Line 54: Replace Kaae Sønder [13] with Sønderby et al. [13].
3. Line 70: replace espite with Despite.
4. Line 80: replace an with the.
5. Line 144: what does the word "excels" mean?
6. Line 209: Replace the comma with a dot.
7. Line 245: et al [26]. replace with et al. [26].
8. Line 260: Are all parentheses correct in [3, 5)?
9. Line 327: replace evaluatedion with evaluation.
10. Line 331: replace notinge with note.
11. Line 333: replace pereformance with performance.
12. Line 356: replace visualizeation with visualization.
13. Lines 380, 385: replace showcases with shows. Same on line 406.
14. Line 386: replace rows with row.
15. Line 406: replace five with four.
16. Line 409, Figure 11: replace hko-7 with HKO-7 in the figure caption. The first panel is missing time step 1 on the x-axis.
17. Line 436: It is replaced by it is.

Author Response
Dear Reviewer,
I hope this letter finds you well. I am writing to submit a revised version of our manuscript titled "Integrating Spatio-Temporal and Generative Adversarial Networks for Enhanced Nowcasting Performance" originally submitted to Remote Sensing. We sincerely appreciate your comprehensive review and appreciate your suggestions for our work.
We would like to extend our gratitude to you for their valuable feedback. You raised concerns regarding certain proprietary terms, a few figures, and identified several spelling and punctuation errors. We have carefully considered each of you suggestions and have made the necessary revisions to address your concerns. Below, we summarize the major changes we made in response to your feedback:
Main comments:
- Line 280: what is dBZ in the pixel expression?
Answer:Thank you for your kind suggestions. In this article, we have made the following modifications to it:
Lines 285-286, dBZ refers to the radar reflectivity factor primarily used to describe the intensity of radar echoes.
- Lines 306-313: is there any physical interpretation of the abbreviations TP, FN, FP, TN?
Answer: Thank you for your kind suggestions. In this article, we have made the following modifications to it:
Lines 334-339, we have included the relevant definitions for TP (True Positive), FN (False Negative), FP (False Positive), and TN (True Negative) to provide readers with a better understanding.
- Line 344, Figure 8: would you like to clarify how the 4 frames of the sequence differ for a specific GAN model?
Answer: Thank you for your kind suggestions. In this article, we have made the following modifications to it:
Lines 374-376, we have provided an explanation for Figure 8.
- Line 389: Figure 10: How does this figure illustrate image prediction? Most likely, there is a good image recognition here?
Answer: Thank you for your kind suggestions. In this article, we have made the following modifications to it:
Lines 412-439, we have conducted a detailed comparison of various figures, adding timestamps to each image to facilitate a better understanding of why the ConvLSTM-TransGAN model performs better in predicting images.
- Line 439, Table 4: what can explain the worsening of indicators for α=0.00001?
Answer: Thank you for your kind suggestions. In this article, we have made the following modifications to it:
Lines 509-512, we have provided an explanation for the worsening of indicators for α=0.00001. The main reason is that when the low Learning rate of 0.00001 is used, the training process of the model will be hindered by falling into the local optimum. This phenomenon prolongs training time, thereby hindering the improvement rate of evaluation indicators and leading to suboptimal results.
Some inaccuracies:
- Line 42: replace learnsing with learning.
Answer: Thank you for your suggestion. In line 42 of this article, we made modifications to it.
- Line 54: Replace Kaae Sønder [13] with Sønderby et al. [13].
Answer: Thank you for your suggestion. In line 54 of this article, we made modifications to it.
- Line 70: replace espite with Despite.
Answer: Thank you for your suggestion. In line 70 of this article, we made modifications to it.
- Line 80: replace an with the.
Answer: Thank you for your suggestion. In line 82 of this article, we made modifications to it.
- Line 144: what does the word "excels" mean?
Answer: Thank you for your suggestion. In line 145 of this article, 'Excels at' means to excel in a certain aspect. In this article, it is pointed out that the ConvLSTM model excels in capturing spatial relationships. To avoid misunderstandings, we have replaced the word for better understanding.
- Line 209: Replace the comma with a dot.
Answer: Thank you for your suggestion. In line 212 of this article, we made modifications to it.
- Line 245: et al [26]. replace with et al. [26].
Answer: Thank you for your suggestion. In line 249 of this article, we made modifications to it.
- Line 260: Are all parentheses correct in [3, 5)?
Answer: Thank you for your suggestion. In line 264 of this article, All parentheses are used correctly.
- Line 327: replace evaluatedion with evaluation.
Answer: Thank you for your suggestion. In line 358 of this article, we made modifications to it.
- Line 331: replace notinge with note.
Answer: Thank you for your suggestion. In line 362 of this article, we made modifications to it.
- Line 333: replace pereformance with performance.
Answer: Thank you for your suggestion. In line 364 of this article, we made modifications to it.
- Line 356: replace visualizeation with visualization.
Answer: Thank you for your suggestion. In line 389 of this article, we made modifications to it.
- Lines 380, 385: replace showcases with shows. Same on line 406.
Answer: Thank you for your suggestion. In line 411, line415 and line 448 of this article, we made modifications to it.
- Line 386: replace rows with row.
Answer: Thank you for your suggestion. In line 421, The plural form should be used here, referring to the following 5 rows.
- Line 406: replace five with four.
Answer: Thank you for your suggestion. In line 448 of this article, we made modifications to it.
- Line 409, Figure 11: replace hko-7 with HKO-7 in the figure caption. The first panel is missing time step 1 on the x-axis.
Answer: Thank you for your suggestion. In Figure 11 of this article, we made modifications to it.
- Line 436: It is replaced by it is.
Answer: Thank you for your suggestion. In line 506 of this article, we made modifications to it.
We conducted a thorough proofreading of the entire manuscript to rectify any spelling and punctuation errors.
We are confident that these revisions have addressed your concerns and significantly improved the manuscript's clarity and accuracy.
Furthermore, we have conducted a comprehensive review of the manuscript to identify and correct any remaining language issues. Additionally, we have carefully adhered to the journal's guidelines and formatting requirements.
Please find attached the revised manuscript along with a point-by-point response to your comments. We extend our gratitude to you for your diligent assessment and insightful feedback, which has been invaluable in refining our research.
Thank you for your time and consideration.
Best regards,
All authors

Reviewer 4 Report
Review of “Integrating Spatio-Temporal and Generative Adversarial Networks for Enhanced Nowcasting Performance” by Wenbin Yu, Suxun Wang, Yadang Chen, Chengjun Zhang, Xinyu Sheng, Yu Yao, Jie Liu, and Gaoping Liu.
The study presents a deep learning methodology to forecast short-term precipitation based on radar data. The authors presented the performance evaluation of ConvLSTM model when coupled with input from TransGAN generator. The statistical performance evaluation metrics presented show that TransGAN-ConvLSTM coupling was capable of predicting short-term features.
Overall, the manuscript needs some work. The figures were fine and legible. I have some major concerns which are listed below. The study is of interest to the scientific community; however, the present manuscript needs additional analysis or at least justification for why this methodology was chosen.
Major comments:
Comment #1: The methodology needs a bit more justification. Authors have compared other models with the TransGAN coupled ConvLSTM. Why not couple the same TransGAN generator to PredRNN or TrajGRU. As shown in Figure 9, ConvLSTM performed poorly after t=2 or 3, but it performed better with TransGAN coupling. However, PredRNN and PredRNN++ performed better compared to ConvLSTM. Along those lines, how would the PredRNN-TransGAN model perform? If coupling the TransGAN generator to PredRNN model is not possible or feasible, then authors need to provide some background on why it is so.
Comment #2: Authors did a good job of testing the sensitivity of the methodology to the learning rate parameter. Why not do the same thing for the other models compared to the study?
Minor comments:
Comment #1: Section 2.1 needs references for the statements given.
Comment #2: Figure 2 is not referred to in the text.
Comment #3: What is FC-LSTM? Make sure to use full forms for abbreviations at their first occurrence in the text.
Comment #4: Equations (1)-(5), define all the symbols and variables. Also, the Hadamard product symbol in equations (1)-(5) was not properly written. In some places, it is used in superscript. Please check. Similarly, define terms used in equation 6.
Comment #5: Line 164, Reference number missing for Shi et al.
Comment #6: Line 203-204: Feels misplaced? What do you mean by “This revised sentence provides a more…”?
Comment #7: Line 210: Define H and W dimensions with respect to the input image.
Comment #8: Line 281: What additional filtering techniques?
English spelling check needs to be done to rectify several misspelled words or sentences.
Author Response
Dear Reviewer,
I hope this letter finds you well. We sincerely appreciate the time and effort you have invested in reviewing our manuscript entitled " Integrating Spatio-Temporal and Generative Adversarial Networks for Enhanced Nowcasting Performance " for Remote Sensing. Your insightful comments and constructive feedback have been immensely valuable in enhancing the quality and rigor of our research.
We are grateful for the positive feedback on our work and are particularly thankful for the constructive questions and suggestions you raised regarding the experimental validation of our proposed model. Your valuable input prompted us to conduct an additional model comparison experiment, which has led to compelling results that strengthen the credibility of our findings.
Specifically, we have incorporated the following changes into the revised manuscript in response to your feedback.
Major comments:
Comment #1: The methodology needs a bit more justification. Authors have compared other models with the TransGAN coupled ConvLSTM. Why not couple the same TransGAN generator to PredRNN or TrajGRU. As shown in Figure 9, ConvLSTM performed poorly after t=2 or 3, but it performed better with TransGAN coupling. However, PredRNN and PredRNN++ performed better compared to ConvLSTM. Along those lines, how would the PredRNN-TransGAN model perform? If coupling the TransGAN generator to PredRNN model is not possible or feasible, then authors need to provide some background on why it is so.
Answer: Thank you for your kind suggestions. In fact, the ConvLSTM model, as a milestone in deep learning for nowcasting, is highly representative, and it already possesses the ability to capture spatio-temporal information. Subsequent models like PredRNN, PredRNN++, etc., are all based on improvements to the ConvLSTM model. The TransGAN generative model optimizes information extraction effectively for these models.
In this experiment, after incorporating the TransGAN image generation model at the output of both ConvLSTM and PredRNN, the results showed little difference between the two. Compared to improved models with more complex structures such as PredRNN, the ConvLSTM model has a simpler network structure and therefore has an advantage in inference performance. Therefore, we chose ConvLSTM as the prediction model in this study.
In this article, we have made the following modifications to it:
In section 5.3, lines 470-493, we conducted experiments and interpretations on this.
Comment #2: Authors did a good job of testing the sensitivity of the methodology to the learning rate parameter. Why not do the same thing for the other models compared to the study?
Answer: Thank you for your kind suggestions. In fact, this paper focuses on optimizing the Learning rate of the proposed model, rather than optimizing the parameters of each classic model. In this paper, the ConvLSTM, TrajGRU, PredRNN, PredRNN++models compared with ConvLSTM-TransGAN have been optimized to the best performance in previous work. We compare these classic models with ConvLSTM-TransGAN models on the basis of achieving the best Learning rate, and get the relevant comparison results.
Minor comments:
Comment #1: Section 2.1 needs references for the statements given.
Answer: Thank you for your kind suggestions. In this article, we have made the following modifications to it:
In section 2.1 We have added citations from references.
Comment #2: Figure 2 is not referred to in the text.
Answer: Thank you for your kind suggestions. In this article, we have made the following modifications to it:
In lines 149-150, we have added reference to Figure 2.
Comment #3: What is FC-LSTM? Make sure to use full forms for abbreviations at their first occurrence in the text.
Answer: Thank you for your kind suggestions. In this article, we have made the following modifications to it:
In lines 147-148, we have added the full forms of FC-LSTM.
Comment #4: Equations (1)-(5), define all the symbols and variables. Also, the Hadamard product symbol in equations (1)-(5) was not properly written. In some places, it is used in superscript. Please check. Similarly, define terms used in equation 6.
Answer: Thank you for your kind suggestions. In lines 158-161, we have made the following modifications to it.
Comment #5: Line 164, Reference number missing for Shi et al.
Answer: Thank you for your kind suggestions. In line 167, we have added citation from references.
Comment #6: Line 203-204: Feels misplaced? What do you mean by “This revised sentence provides a more…”?
Answer: Thank you for your kind suggestions. We have removed this sentence to make the article more concise and clear.
Comment #7: Line 210: Define H and W dimensions with respect to the input image.
Answer: Thank you for your kind suggestions. In line 213, we explained H and W.
Comment #8: Line 281: What additional filtering techniques?
Answer: Thank you for your kind suggestions. In section 4.2, lines 295-311, we added the process of data preprocessing.
We would like to express our sincere gratitude for guiding us towards these improvements, which have elevated the quality and significance of our work. Your expertise and thorough evaluation have undoubtedly been instrumental in shaping the manuscript.
Thank you once again for your dedication and valuable contributions to our manuscript. We greatly appreciate your time and expertise. Your feedback has been invaluable in enhancing the rigor and impact of our research.
Thank you for your time and consideration.
Best regards,
All authors

Round 2
Reviewer 4 Report
I would like to thank the authors for addressing the reviewer's suggestions, especially for adding Table 4 and related discussion. The manuscript has improved a lot and is now acceptable for publication.